evolution, genetics, genomics

somatic mutation, bioinformatics, plants, mutation rate

**Authors for correspondence:**
Reed A. Cartwright
e-mail: cartwright@asu.edu
Robert Lanfear
e-mail: rob.lanfear@anu.edu.au

†Equal contribution.

# A phylogenomic approach reveals a low somatic mutation rate in a long-lived plant

Adam J. Orr[1,†], Amanda Padovan[2,3,†], David Kainer[2,†], Carsten Külheim[2,4], Lindell Bromham[2], Carlos Bustos-Segura[2], William Foley[2], Tonya Haff[2], Ji-Fan Hsieh[2], Alejandro Morales-Suarez[5], Reed A. Cartwright[1] and Robert Lanfear[2,5]

[1]The Biodesign Institute and the School of Life Sciences, Arizona State University, Tempe, AZ, USA
[2]Division of Ecology and Evolution, Research School of Biology, Australian National University, Canberra, Australia
[3]CSIRO Black Mountain Science and Innovation Park, Canberra, ACT 2601, Australia
[4]School of Forest Resources and Environmental Science, Michigan Technological University, Houghton, MI 49931, USA
[5]Department of Biological Sciences, Macquarie University, Sydney, Australia

LB, 0000-0003-2202-2609; RL, 0000-0002-1140-2596

Somatic mutations can have important effects on the life history, ecology and evolution of plants, but the rate at which they accumulate is poorly understood and difficult to measure directly. Here, we develop a method to measure somatic mutations in individual plants and use it to estimate the somatic mutation rate in a large, long-lived, phenotypically mosaic *Eucalyptus melliodora* tree. Despite being 100 times larger than *Arabidopsis*, this tree has a per-generation mutation rate only ten times greater, which suggests that this species may have evolved mechanisms to reduce the mutation rate per unit of growth. This adds to a growing body of evidence that illuminates the correlated evolutionary shifts in mutation rate and life history in plants.

## 1. Background

Trees grow from multiple meristems which contain stem cells that divide to produce the somatic and reproductive tissues. A mutation occurring in a meristematic cell will be passed on to all resulting tissues, potentially causing an entire branch including leaves, stems, flowers, seeds and pollen to have a genotype different from the rest of the plant [1,2]. These different genotypes may lead to phenotypic changes, potentially with important consequences for plant ecology and evolution [3–8]. For example, somatic mutations could explain how long-lived plants adapt to changing ecological conditions [9], and are thought to influence long-term variation in the rates of evolution and speciation among plant lineages [10]. Somatic mutations can degrade genetic stocks used in agriculture and forestry [11,12], confer herbicide resistance to weed species [13] and have been linked to declining plant fitness in polluted areas [14]. However, despite the importance of somatic mutations and recent progress in understanding them [1,2,15–18], there remain significant analytical challenges in inferring somatic mutation rates from sequencing data in plants.

We present a solution to the challenges of measuring the somatic mutation rate that leverages the phylogeny-like structure of the plant itself to estimate the genome-wide somatic mutation rate of the individual. Our strategy has three key features. First, we sequence the full genome of three biological replicates of eight branch tips. Using three biological replicates per branch tip significantly reduces the false-positive rate, because many types of error (e.g. sequencing error or mutations induced during DNA extraction or library

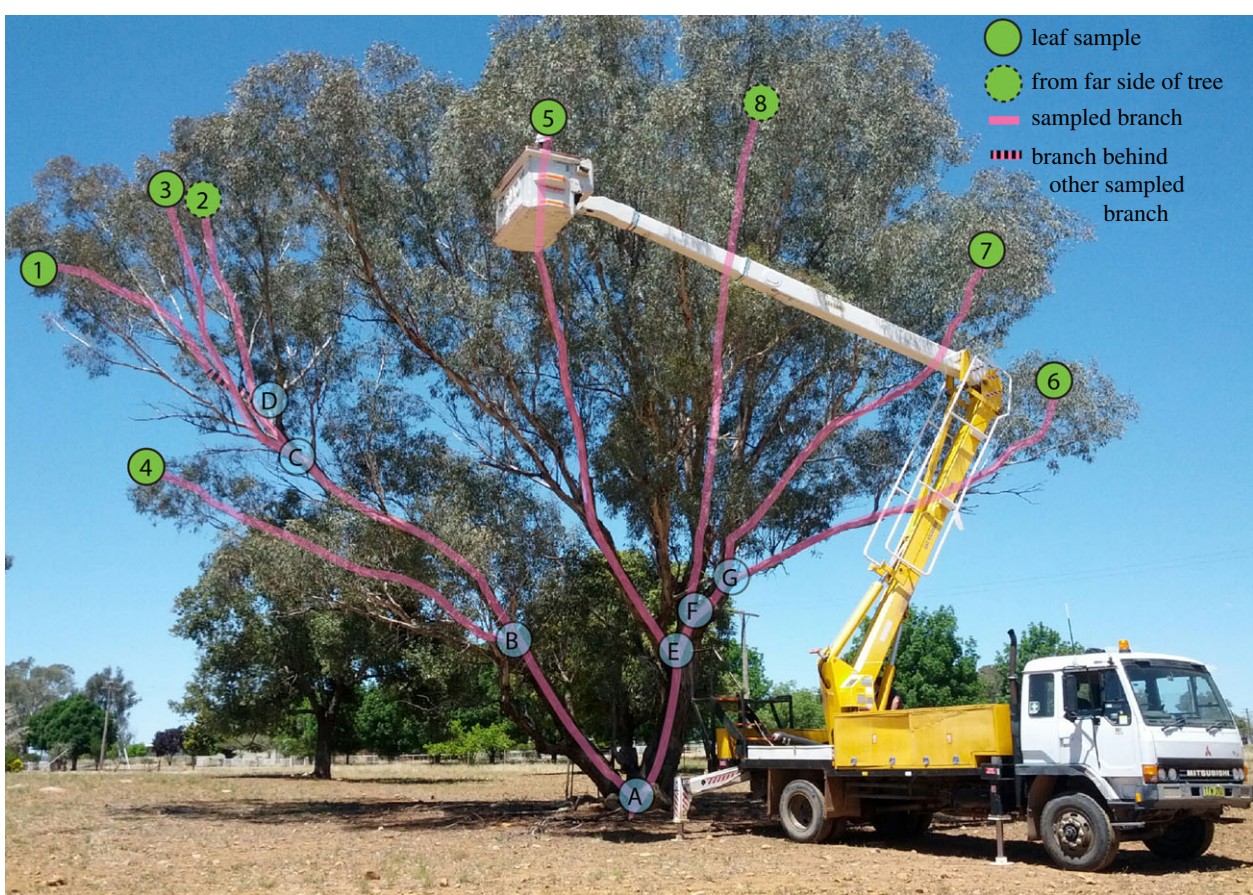

**Figure 1.** The *Eucalyptus melliodora* individual sequenced in this study. The eight branch tips sampled are shown by numbered green circles with internal nodes of the tree shown as letters in blue circles. Circles with dashed outlines are from the far side of the tree. Pink lines trace the physical branches that connect the sampled tips. The herbivore-resistant branch comprises samples 1–3.

preparation) are very unlikely to appear at the same position in all three replicates, making it easy to distinguish these errors from biological signal. Second, our strategy includes an inbuilt positive control, because we can ask whether the phylogenetic tree we reconstruct from the set of putative somatic mutations across the eight branch tips reflects the known physical structure of the tree (i.e. whether phylogeny correctly reconstructs ontogeny, as is expected for plant development in most cases, but see below). Third, the approach allows us to estimate the false-negative rate and the false-discovery rate of our inferences directly from the replicate samples (see below).

We applied this approach to a long-lived yellow box (*Eucalyptus melliodora*) tree, notable for its phenotypic mosaicism: a single large branch in this individual is resistant to defoliation by Christmas beetles (*Anoplognathus* spp Coleoptera: Scarabaeidae) due to stable differences in leaf chemistry and gene expression [19,20]. We find that the rate of somatic mutation per generation is relatively high, but the rate per metre of growth is surprisingly low in comparison to other species. We suggest potential proximate and ultimate reasons for the wide variation in somatic mutation rates across plants.

## 2. Material and methods

### (a) Field sampling
We used a known mosaic *E. melliodora* (yellow box). This tree is found near Yeoval, NSW, Australia (−32.75°, 148.65°). We collected the ends of eight branches in the canopy (figure 1). Branches were collected using an elevated platform mounted

on a truck and were placed into labelled and sealed polyethylene bags which were immediately buried in dry ice in the field. Within the 24 h of collection, the samples were transferred to −80°C until DNA extraction. Simultaneously, we used a thin rope to trace each branch from the tip to the main stem. These rope lengths were measured to determine the lengths of the physical branches of the tree.

### (b) DNA extraction, library preparation and sequencing
The branches were maintained below −80°C on dry ice and in liquid nitrogen while sub-sampled in the laboratory. From each branch, we selected a branch tip which had at least three consecutive leaves still attached to the stem. From this branch tip, we independently sub-sampled roughly 100 mg of leaf from the 'tip-side' of the mid-vein on three consecutive leaves using a single hole punch into a labelled microcentrifuge tube containing two 3.5 mm tungsten carbide beads. The sealed tube was submerged in liquid nitrogen before the leaf material was ground in a Qiagen TissueLyser (Qiagen, Venlo, Netherlands) at 30 Hz in 30 s intervals before being submerged in liquid nitrogen again. This was repeated until the leaf tissue was a consistent powder, up to a total of 3.5 min grinding time.

DNA was extracted from this leaf powder using the Qiagen DNeasy Plant Mini Kit (Qiagen, Venlo, The Netherlands), following the manufacturer's instructions. DNA was eluted in 100 µl of elution buffer. DNA quality was assessed by gel electrophoresis (1% agarose in 1 × TAE containing ethidium bromide), and quantity was determined by Qubit Fluorometry (Invitrogen, California, USA) following manufacturer's instructions.

We used a Bioruptor (Diagnode, Seraing (Ougrée), Belgium) to fragment 1 µg of DNA to an average size of 300 bp (35 s on 'High', 30 s off for 35 cycles at 4°C). The fragmented DNA was

*Proc. R. Soc. B* **287**: 20192364

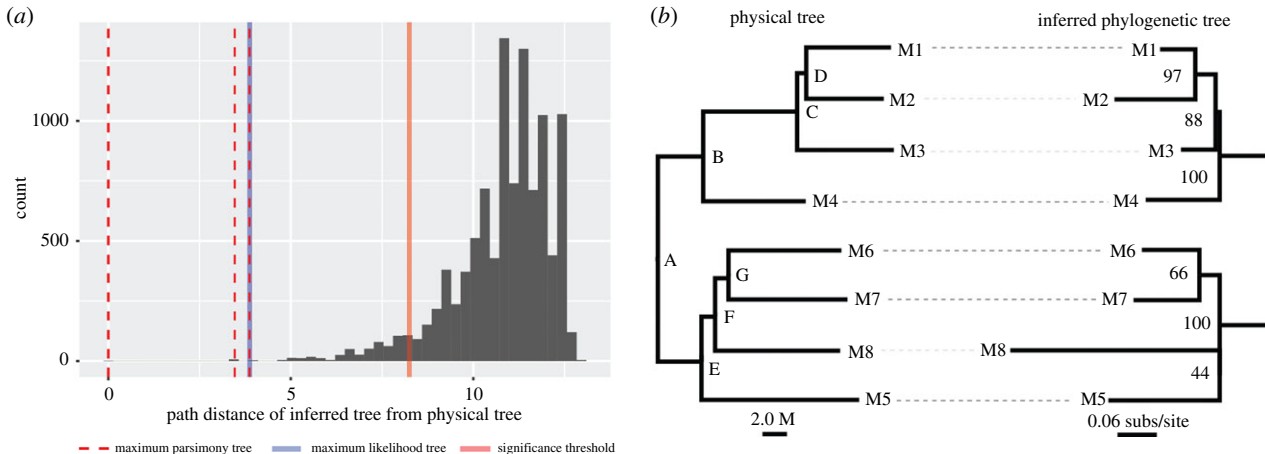

**Figure 2.** Phylogenetic trees reconstructed from somatic mutations resemble the physical structure of the tree more closely than expected by chance. (*a*) The PD between the physical tree (figure 1) and all 10 395 possible phylogenetic trees of eight taxa is shown as a histogram. A tree with the same topology as the physical tree will have a PD of 0. The solid red line represents the boundary of the smallest 5% of the distribution of PDs, such that a tree with a PD lower than this line is more similar to the physical tree than expected by chance. All of the maximum-parsimony trees (dashed red lines) and the one maximum-likelihood tree (solid blue line) are more similar to the physical tree than expected by chance. (*b*) A side-by-side comparison of the physical tree (left, branch lengths in metres) and the maximum-likelihood tree (right, branch lengths in substitutions per site) inferred with the JC model. Letters on the nodes of the physical tree (left) correspond to the same letters of internal nodes in figure 1. Numbers on the maximum-likelihood tree (right) are bootstrap percentages. There is a single difference between the two trees: the inferred tree groups samples M8 and M5 together with low bootstrap support (44%), which is a grouping that does not occur in the physical tree.

purified using 1.6 × SeraMag Magnetic Beads (GE LifeSciences, Illinois, USA) following the manufacturer's instructions. We used Illumina TruSeq DNA Sample Preparation kit (Illumina Inc., California, USA) following the manufacturer's instructions to generate paired-end libraries for sequencing. These libraries were sequenced on an Illumina HiSeq 2500 (Illumina Inc., California, USA) at the Biomolecular Resource Facility at the Australian National University, Canberra.

## (c) Creation of pseudo-reference genome

Since there is no available reference genome for *E. melliodora*, we created a pseudo-reference genome by iterative mapping and consensus calling. To do this, we first mapped all of our reads to version 2.1 of the *E. grandis* reference genome [21] using NGM [22] and then updated the *E. grandis* reference genome using bcftools consensus [23]. We iteratively repeated this procedure until we saw only marginal improvement in the number of unmapped reads and reads that mapped with a mapping quality of zero. The alignment originally contained 67 M unmapped reads and 311 M reads that mapped with zero mapping quality, out of a total of 1792 M reads. After the first iteration, the alignment contained 61 M unmapped reads and 349 M reads that mapped with zero mapping quality. After the last iteration, the alignment contained 59 M unmapped reads and 311 M reads that mapped with zero mapping quality. The consensus of this alignment served as the reference for all further downstream analyses.

## (d) Variant calling for positive control

To call variants for the positive control, we mapped each replicate of each branch tip (24 samples in total) to the final pseudo-reference genome using NGM and called genotypes using GATK 4 according to the GATK best practices workflow [24]. This resulted in a full genome alignment of all 24 samples (three replicates of eight branches) and produced an initial set of 9 679 544 potential variable sites, a number which includes all heterozygous sites in the genome.

We then filtered variants to minimize the false-positive rate by retaining only those sites in which: (i) genotype calls were identical within all three replicates of each branch tip (see also electronic

supplementary material, §1); (ii) at least one branch tip had a different genotype than the other branch tips; (iii) the site is biallelic, since multiple somatic mutations are likely to be extremely rare; (iv) the total depth across all samples is less than or equal to 500 (i.e. roughly twice the expected depth of 240×), since excessive depth is a signal of alignment issues; (v) the ExcessHet annotation was less than or equal to 40, since excessive heterozygosity at a site is a sign of genotyping errors, particularly in a site that is actually uniformly heterozygous throughout the tree but at which genotyping errors have caused a mutation to be called; and (vi) the site is not in a repetitive region determined by a lift-over of the *E. grandis* RepeatMasker annotation, as variation in repeat regions is often due to alignment error. This filtering produced a set of 99 high-confidence sites containing putative somatic mutations. The number of mutations that remained after the application of each filter is described in §5 of the electronic supplementary material.

## (e) Positive control

Using the set of 99 high-confidence putative somatic mutations, we use the Phangorn package in R [25] to calculate the parsimony score of all 10 395 possible phylogenetic trees of eight taxa. This estimates the number of somatic mutations that would be required to explain each of the 10 395 phylogenetic trees, using the Fitch algorithm implemented in the Phangorn package. Of these trees, three had the maximum-parsimony score of 78. One of these three trees matched the topology of the physical tree (figure 2).

Next, we calculated the path difference (PD) between all 10 395 trees and the physical tree topology. The PD measures differences between two phylogenetic tree topologies [26] by comparing the differences between the path lengths of all pairs of taxa. Here we use the variant of the PD that treats all branch lengths as equal, because we are interested only in topological differences between trees, not branch length differences. Comparing all 10 395 trees to the physical tree topology provides a null distribution of PDs between all trees and the physical tree topology, which we can use to ask whether each of the three maximum-parsimony trees is more similar to the physical tree topology than would be expected by chance. To do this, we simply ask whether the PD of each of the three observed maximum-parsimony trees falls within the lower 5% of the

distribution of PDs from all 10 395 trees. This was the case for all three maximum-parsimony trees ($p < 0.001$ in all cases; figure 2), suggesting that our data contain biological signal which render the phylogenetic trees reconstructed from somatic mutations more similar than would be expected by chance to the physical tree.

## (f) Variant calling for estimating the rate and spectrum of somatic mutations

Using the physical tree topology to define the relationship between samples, we called somatic mutations using DeNovo-Gear's dng-call method [27] compiled from https://github.com/denovogear/denovogear/tree/3ae70ba. Model parameters were estimated from 3-fold degenerate sites in our NGM alignment, via VCFs generated by bcftools mpileup and bcftools call with–pval-threshold = 0. We estimated maximum-likelihood parameters using the Nelder–Mead numerical optimization algorithm implemented in the R package dfoptim (https://cran.r-project.org/package=dfoptim). We then called genotypes using the GATK best practices workflow as above, but with the standard-min-confidence-threshold-for-calling argument set to 0, causing the output VCF to contain every potentially variable site in the alignment. Thus, we used GATK to generate high-quality pileups from our alignments. These pileups were then analysed by dng-call to identify (i) heterozygous sites and (ii) de novo somatic mutations. Since successful haplotype construction in a region indicates a high-quality alignment, we used Whatshap 0.16 [28] to generate haplotype blocks from the heterozygous sites.

Next, we filtered our de novo variant set to remove potential false positives. We removed variants that (i) were on a haplotype block with a size less than 500 nucleotides (among other things, this filter will remove many putative variants that fall in long repeat regions); (ii) were within 1000 nucleotides of another de novo variant (indicative of alignment issues such as might occur in repeats and other regions); (iii) had an log likelihood of the data (LLD) score less than −5 (indicative of poor model fit); and (iv) had a de novo mutation probability (DNP) score less than 0.99999 (retaining only the highest confidence variants). This produced a final variant set of 90 variants.

## (g) Estimation of the false-negative rate

To estimate the number of mutations that we were likely to have filtered out in our variant calling pipeline, we used the method of Ness *et al.* [29], adapted to the current phylogenetic framework. Specifically, we randomly selected 14 000 sites from the first 11 scaffolds of the pseudo-reference genome and randomly assigned 1000 of these sites to each of the 14 branches on the tree. For each of these sites, we induced *in silico* mutations into the raw reads with a three-step procedure. We first estimated the observed genotype at the root using DeNovoGear call at each site. We then chose a mutant genotype by mutating one of the alleles to a randomly chosen different base using a transition/transversion ratio of 2, reflecting the observed transition/transversion ratio of eucalypts. We edited the raw reads as follows: for each mutation, we defined the samples to be mutated as all of those samples that descend from the branch on which the *in silico* mutation occurred. For example, an *in silico* mutation occurring on branch B → C in figure 1 would affect all three replicates of samples 1, 2 and 3. We then edited the reads that align to the site in question to reflect the new mutation, depending on whether the reference genotype was homozygous or heterozygous. For homozygous sites, we selected the number of reads to mutate by generating a binomially distributed random number with a probability of 0.5 and a number of observations equal to the number of reads with the reference genotype. We then randomly selected that the number of reads with the reference allele to mutate to the mutant allele

and edited the raw reads accordingly. For a heterozygous site, we edited the reads to replace all occurrences of the reference allele to mutant allele. The result of this procedure is the generation of a new set of raw fastq files, which now contain information on 1000 *in silico* mutations for every branch in the physical tree.

To determine the false-negative rate of the variant calling pipeline, we re-ran the entire pipeline using the edited reads and recorded how many of the 14 000 *in silico* mutations were recovered by the pipeline. This number was 4193, suggesting that our false-negative rate is 70.05%. In other words, we expect that our empirical analysis recovered roughly three in 10 true mutations, because our power is limited in part by attempts to filter out false positives, which also removes a number of true positives.

## (h) Estimation of the false-discovery rate

To determine the false-discovery rate of the variant calling pipeline, we simulated random trees of our samples (where each of the eight branches is represented by three tips that denote the three replicates of that branch) by shuffling the tip labels until the tree had a maximal Robinson–Foulds distance from the original tree. This 24-taxon tree shares no splits with the original 24-taxon tree, so any phylogenetic information should be removed. We simulated 100 such trees and called variants using the pipeline above, but assuming that these trees were the physical tree, and ignoring any sites we had previously called as variable. Thus, any variants called by the pipeline must be false positives. We recovered 11 false positive calls over 100 simulations (i.e. 0.11 false-positive mutations per simulation), indicating our false-discovery rate is approximately 0.12%. We calculated the false-discovery rate only once, after the details of the pipeline were finalized, to avoid overfitting our pipeline to artefactually reduce the false-discovery rate.

# 3. Results and discussion

## (a) Field sampling and sequencing

We selected eight branch tips that maximized the intervening physical branch length on the tree (figure 1), reasoning that this would increase our power by maximizing the number of sampled cell divisions and thus somatic mutations. We performed independent DNA extractions from three leaves from each branch tip, prepared three independent libraries for Illumina sequencing and sequenced each library to 10× coverage (assuming a roughly 500 Mbp genome size, as is commonly observed in *Eucalyptus* species [30]) using 100 bp paired-end sequencing on an Illumina HiSeq 2500. Quality control of the sequence data verified that each sample was sequenced to approximately 10× coverage and that each branch tip was therefore sequenced to approximately 30× coverage.

## (b) Positive control analysis

We first performed a positive control to confirm that the phylogeny of a set of high-confidence somatic variants matches the physical structure of the tree. This approach relies on being able to infer the ontogeny of the tree with sufficient accuracy that a valid comparison can be made between the ontogeny of the tree and a phylogeny generated from that tree's somatic variants. Documenting a plant's ontogeny with sufficient accuracy may not be possible for all plant species or individuals. Nevertheless, the physical structure of the tree we studied was clear (figure 1), and although *Eucalyptus* trees are known to frequently lose branches, branch loss and regrowth should not affect the correlation between

ontogeny and phylogeny provided that sufficient mutations accumulate during cell replication. To perform the phylogenetic positive control, we created a pseudo-reference genome using our data to update the genome of *E. grandis* (see methods). We then called variants using GATK [31] in all three replicates of all eight branch tips and used a set of strict filters (see methods and supplementary information) designed to avoid false-positive mutations in order to arrive at an alignment of 99 high-confidence somatic variants. To find the phylogenetic trees that best explain this alignment, we calculated the alignment's parsimony score on all 10 395 possible phylogenetic trees of eight samples. Parsimony is an appropriate method here because we do not expect more than one mutation to occur at any single site on any single branch of the *E. melliodora* tree. We then asked whether the three phylogenetic trees with the most parsimonious scores were more similar to the physical structure of the tree than would be expected by chance. To do this, we calculated the PD between the structure of the physical tree and each of the three most parsimonious trees. We then compared these differences to the null distribution of PDs generated by comparing the structure of the physical tree to all possible 10 395 trees of eight samples (figure 2a). All three maximum-parsimony trees were significantly more similar to the physical tree than would be expected by chance ($p < 0.001$ in all cases; figure 2a, dashed red lines). Furthermore, one of the most parsimonious trees is identical to the structure of the physical tree, and a maximum-likelihood tree calculated from the same data shows just one topological difference compared to the structure of the physical tree, in which sample 8 is incorrectly placed as sister to sample 5, but with low bootstrap support of 44% (figure 2a, blue line; figure 2b). As would be expected if plants accumulate somatic mutations as they grow, there is a significant correlation between the branch lengths of the physical tree measured in metres and the branch lengths of the maximum-parsimony tree of the same topology measured in number of somatic mutations (linear model forced through the origin: $R^2 = 0.82$, $p < 0.001$; see also electronic supplementary material, §4). Notably, while various factors such as the difficulty of correctly inferring plant ontogeny may limit the utility of a phylogenetic positive control such as we present here (i.e. may produce false-negative results in which the structure of the tree appears, erroneously, to differ from the phylogeny of the sequenced genomes), it is unlikely that these factors would erroneously cause a close match between the physical structure of the tree and a phylogeny generated from the genomes of eight branches of that tree (i.e. a false positive). We therefore conclude that these analyses demonstrate that the phylogeny recovered from the genomic data matches the physical structure of the tree and confirm that there is a strong biological signal in our data.

## (c) Estimation of the somatic mutation rate

We next developed a full maximum-likelihood framework that extends the existing models in DeNovoGear [27] to detect somatic mutations in a phylogenetic context and used this framework to estimate the full rate and spectrum of somatic mutations in the individual *E. melliodora* (see Material and methods). This method improves on the approach we used in our positive control, above, because it increases our power to detect true somatic mutations and avoid false positives by

assuming that the phylogenetic structure of the samples follows the physical structure of the tree, an assumption that is validated by the analyses above. It also makes better use of the replicate sampling design than the method we use for our positive control, above, by directly modelling the expected variation in sequencing data across our three biological replicates under the expectation that all three replicates were sequenced from a single underlying genotype (see methods and electronic supplementary material). Using this framework, we identified 90 high-confidence somatic variants.

Of the 90 high-confidence variants we identified, 20 were in genes. Of these, six were in coding regions, with five non-synonymous mutations and one synonymous mutation. The small sample size of synonymous and non-synonymous mutations means that we cannot provide a meaningful estimate of the ratio of non-synonymous to synonymous somatic mutations, although such an estimate would help to understand the extent to which somatic mutations may be under selection. We detected seven mutations on the branch that separates the herbivore-resistant samples from the other samples (branch B → C, figure 1). Although we lack the functional evidence to determine whether any of these mutations are directly involved in the resistance phenotype, two of the mutations occur near genes that are plausible candidates for further investigation. One mutation occurs near Eucgr.C00081, which is a zinc-binding CCHC-type protein belonging to a small protein family known to bind RNA or ssDNA in *Arabidopsis thaliana* and thus potentially involved in gene expression regulation. Another mutation occurs near Eucgr.I01302, an acid phosphatase that may have as a substrate phosphoenol pyruvate, and therefore may be involved in pathways associated with the production of various secondary metabolites, including those identified in a recent GWAS study in a closely related eucalypt [32].

We used the replicate sampling design of our analysis to estimate the false-negative rate and the false-discovery rate of our approach. It is necessary to estimate both the number of false-negative mutations and the number of false-positive mutations in order to estimate a somatic mutation rate. The former allows one to correct for the number of somatic mutations which a pipeline has failed to detect, while the latter allows one to correct for the number of somatic mutations which a pipeline has erroneously inferred. We estimated the false-negative rate by creating 14 000 *in silico* somatic mutations in the raw reads [33], comprising 1000 *in silico* mutations for each of the 14 branches of the physical tree, and measuring the recovery rate of these *in silico* mutations using our maximum-likelihood approach. We were able to recover 4193 of the *in silico* mutations, suggesting that our recovery rate is 29.95%, and thus our false-negative rate is 70.05%. This false-negative rate was similar across all of the 14 branches in the tree (see electronic supplementary material, §2). Our ability to recover mutations differs substantially between repeat regions and non-repeat regions: we recover 40% of the simulated mutations in non-repeat regions, but just 13% of the simulated mutations in repeat regions (which make up roughly 40% of the genome). This difference is explained primarily by the stringent filters we use, that lead us to screen out many putative somatic mutations in repeat regions. We then estimated the number of false-positive mutations in our data, and hence the false-discovery rate (the percentage of the observed mutations that are false positives) by repeating our detection pipeline

after permuting the labels of samples and replicates to remove all phylogenetic information in the data, and only considering sites that we had not previously identified as variable (see methods). By removing phylogenetic information and previously identified variable sites, we can be sure that any mutations detected by this pipeline are false positives. Across 100 such permutations, we detected 11 false-positive mutations in total, suggesting that our pipeline generates 0.11 false-positive variant calls per experiment, and that the false-discovery rate for our analysis is 0.12%.

Based on these analyses, we can estimate the mutation rate per metre of physical growth and per year. We estimate that the true number of somatic mutations in our samples is 300 (calculated as: (90 high-confidence mutations minus 0.11 false-positive mutations)/the recovery rate of 0.2995)). Since we sampled a total of 90.1 m of physical branch length, this equates to 3.3 somatic mutations per diploid genome per metre of branch length, or $2.75 \times 10^{-9}$ somatic mutations per base per metre of physical branch length. Although the exact age of this individual is unknown and difficult to estimate—it lives in a temperate climate and does not produce growth rings—its age is nevertheless almost certainly between 50 and 200 years old. Given that the physical branch length connecting each sampled branch tip to the ground varies between 8.4 m and 20.3 m, we estimate that the mutation rate per base per year for a single apical meristem lies in the range $1.16 \times 10^{-10}$ to $1.12 \times 10^{-9}$ (i.e. $8.4 \times 2.75 \times 10^{-9}/200$ to $20.3 \times 2.75 \times 10^{-9}/50$). It is important to note that it remains unclear whether mutations in growing plants accumulate linearly with the amount of physical growth. Indeed, evidence is accumulating that in at least some (and perhaps most) species, mutations may accumulate primarily at branching events rather than during elongation of individual branches [34,35]. If this is the case, then the correlation we observe between the physical branch length and the number of inferred somatic mutations (see above, and electronic supplementary material, §4) may be due to a correlation between the physical length of a branch and the number of branching events that occurred along that branch during the plant's development. It is not possible to directly estimate the number of branching events along each branch in the individual tree we used in this study, because we expect that the tree will have regularly lost branches throughout its life, leaving no accurate record of the number of branching events.

## (d) What drives differences in somatic mutation rates among species?

With some additional assumptions, it is also possible to estimate the mutation rate per generation and to compare this to estimates from other plants. The average height of an adult E. melliodora individual is between 15 m and 30 m [36], so if we assume that all somatic mutations are potentially heritable (about which there is limited evidence [1] and ongoing discussion [37]), we can estimate the per-generation mutation rate. To do this, we assume that a typical seed will be produced from a branch that has experienced 15–30 m of linear growth from the seed [36], and that mutations will have accumulated along that branch at $2.75 \times 10^{-9}$ somatic mutations per base per metre of physical branch length, estimated above. We therefore estimate that the heritable somatic mutation rate per generation is between $4.13 \times 10^{-8}$ and $8.25 \times 10^{-8}$ mutations per base. For comparison the roughly 20 cm tall Arabidopsis thaliana has a per-generation mutation rate of $7.1 \times 10^{-9}$ mutations per base [38]. To the extent that such a comparison is accurate, which will be somewhat limited because the former estimate considers only somatic mutations and the latter considers all heritable mutations including those caused during meiosis, we can then compare these estimates. Comparing the estimates suggests that despite being roughly 100 times taller than Arabidopsis thaliana, the per-generation mutation rate of E. melliodora is just approximately 10 times higher, which is achieved by a roughly fifteen-fold reduction in the mutation rate per physical metre of plant growth.

Our work adds to a growing body of evidence that low somatic mutation rates per unit of growth are a general feature of many large plant species [1,2,15,16,18]. For example, a recent study of the Sitka spruce estimated a per-generation somatic mutation rate of $2.7 \times 10^{-8}$, with confidence intervals that overlap ours [15]. While this per-generation rate is very similar to the one we estimate here, the rate of somatic mutation per metre of growth is around an order of magnitude lower in the Sitka spruce than our estimate for E. melliodora ($2.75 \times 10^{-9}$ somatic mutations per base pair per metre of growth for E. melliodora estimated here, versus $3.5 \times 10^{-10}$ somatic mutations per base pair per metre of growth for Sitka spruce, estimated by dividing the per-generation mutation rate of $2.7 \times 10^{-8}$ mutations per base by the average height of studied individuals of 76 m [15], an appropriate calculation because the somatic mutation rate was estimated from paired samples taken from the base and the top of a collection of individual trees). Lower somatic mutation rates per unit of growth in larger plants may be the result of selection for reduced somatic mutation rates in response to the accumulation of increased genetic load in larger individuals [1,2,10,15,39–41]. This pattern could also explain why larger plants tend to have lower average rates of molecular evolution than their smaller relatives [10,42].

Several possible mechanisms might account for a reduction in accumulation of mutations per unit of growth in larger plants. Selection may favour reduction in the mutation rate per cell division through enhanced DNA repair to reduce the lifetime mutation risk. Alternatively, it may be that the reduction in the mutation rate is due to slower cell division. For example, plant meristems contain a slowly dividing population of cells in the central zone of the apical meristem, and these cells are known to divide more slowly in trees than in smaller plants [43]. Indeed, the rate of cell division in the central zone is so low that one estimate put the total number of cell divisions per generation in large trees as low as one hundred [43]. Regardless of the underlying mechanism, the surprisingly low rates of somatic mutation in large plants reported here and elsewhere suggest an emerging picture in which there is a strong link between the somatic mutation rates and life history across the plant kingdom. Longevity and size are two aspects of plant life history likely to be of central importance to the evolution of somatic mutation rates. Larger plants may suffer from a higher accumulation of somatic mutations because of the necessity for additional cell divisions. Plants that live longer may suffer from a higher accumulation of somatic mutations because of the accumulation of DNA damage over time and/or increased cell turnover in long-lived tissues. The relative importance of these two factors may differ among clades, species and individual tissues and is likely to also depend on the balance between DNA damage and repair

between cell divisions [44], the accuracy of DNA replication, cell size and the rate of cell division. We hope that the approach we describe here will help in further understanding how these and other factors contribute to the accumulation or avoidance of somatic mutations in plants.

Data accessibility. All of the bioinformatic workflows we describe here are provided in full detail in the GitHub repository available at https://github.com/adamjorr/somatic-variation. The raw short-read data are available online at https://www.ncbi.nlm.nih.gov/bioproject/553104.

Authors' contributions. R.L. conceived the study; D.K., A.P., C.K., L.B., W.F., T.H. and R.L. planned the study; A.J.O., D.K., A.M.-S., R.A.C. and R.L. involved in bioinformatic analysis; A.J.O., D.K. and R.A.C. involved in bioinformatic methods development; A.P. helped in laboratory work; A.P., D.K., C.B.-S., T.H. and J.-F.H. involved in fieldwork; R.L.wrote the paper; all authors discussed and interpreted results; All authors edited many drafts of the paper.
Competing interests. We declare we have no competing interests.
Funding. This work was supported by a Hermon-Slade grant to R.L. and an Australian Research Council Future Fellowship to R.L.

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
