## [Reviewer comments · Proceedings of the Royal Society B: Biological Sciences]

Review History

RSPB-2019-2364.R0 (Original submission)

Review form: Reviewer 1 (Dacheng Tian)

Recommendation

Accept with minor revision (please list in comments)

Scientific importance: Is the manuscript an original and important contribution to its field?

Good

General interest: Is the paper of sufficient general interest?

Good

Quality of the paper: Is the overall quality of the paper suitable?

Good

Is the length of the paper justified?

Yes

Should the paper be seen by a specialist statistical reviewer?

No

Do you have any concerns about statistical analyses in this paper? If so, please specify them explicitly in your report.

Yes

It is a condition of publication that authors make their supporting data, code and materials available - either as supplementary material or hosted in an external repository. Please rate, if applicable, the supporting data on the following criteria.

Is it accessible?

Yes

Is it clear?

Yes

Is it adequate?

Yes

Do you have any ethical concerns with this paper?

No

Comments to the Author

I had read this paper in preprint arxiv and in fact I like this paper. So I'm delighted to have the chance to give a formal review on it. The somatic mutations in large long-lived trees are attracting increasing interests from the community, and the current work by the authors could certainly serve as an excellent paradigm to develop related methodologies. Though I would like to see a quick publication of this work, I may still have to be rigorous to ensure this paper to be published as qualified as it could be.

My general concern is about the many assumptions made in this paper, some of which seem to be weak and could limit the application of the proposed method. The details are given below.

The phylogenomic approach is indeed intriguing, but related tests depend on the known physical structure of the tree. The clear ontogeny is not always easy to obtain on many early-branching or multi-stemmed trees. Besides, old large trees are vulnerable to environmental stresses, and old branches could be damaged while new branches could be regenerated near wounded sites. These factors complicate the true ontology as well as the underlying relationship to phylogeny. It's thus not clear whether the phylogenomic approach is robust enough against all these confounding factors.

Page 3 Line 46. Using three biological replicates could certainly reduce many non-context-dependent errors (i.e., errors assumed to be random by position), but not always true for context-dependent errors. Could we check how many false-positives are eliminated by checking if we only require one or two replicates in mutation calling. In turn, is it possible that this may also lead to false negatives as any replicate with poor coverage on certain sites could cause a rejection of true mutations.

Page 7 Line 127. What causes this very high false negative rate here? My guess is that a high divergence with reference genome could lead to fewer regions accessible, and removing repetitive regions also reduce the accessible regions. How many regions are actually valid in variant calling and how many regions are repetitive defined by RepeatMasker then? Or are there other factors lead to this high false negative rate?

Page 7 Line 128. Using simulation to estimate the false positives is nearer to a debugging process,

which means most times if we find these false positives we could re-adjust the criteria to eliminate them. If we could not in fact reject them, even after manual inspection, how can we be sure that they are truly false positives? So how do the 11 false-positive mutations look like? Are they failed any criteria or just not match well with the ontology?

Page 7 Line 136. By calculating per meter mutation rate, the implied assumption here is that mutations fixed linearly with the branch length. I had to say that I don't agree very well on this assumption. There are growing bodies of evidences suggest that the branching processes (e.g., from SAM to axillary meristems) rather than the elongation processes generating most fixed mutations. Considering two branch tips with same lengths to the ground but with different branching orders, the higher-ordered branch tip could have higher per meter mutation rate as it undergoes much more stem cell divisions.

Review form: Reviewer 2

Recommendation

Accept with minor revision (please list in comments)

Scientific importance: Is the manuscript an original and important contribution to its field?

Excellent

General interest: Is the paper of sufficient general interest?

Excellent

Quality of the paper: Is the overall quality of the paper suitable?

Excellent

Is the length of the paper justified?

Yes

Should the paper be seen by a specialist statistical reviewer?

No

Do you have any concerns about statistical analyses in this paper? If so, please specify them explicitly in your report.

No

It is a condition of publication that authors make their supporting data, code and materials available - either as supplementary material or hosted in an external repository. Please rate, if applicable, the supporting data on the following criteria.

Is it accessible?

Yes

Is it clear?

Yes

Is it adequate?

Yes

Do you have any ethical concerns with this paper?

No

Comments to the Author

This is a very interesting, clear, well-written paper on a current hot topic - that of the extent of somatic mutations in plants in general, and trees in particular. The potential contribution of somatic mutations to genetic diversity of long-lived plants has been a topic of theoretical discussion for decades, but only recently has sequencing technology and cost been sufficient to address it empirically. There have been several papers on this topic in the past ~2 years - prior to that, there were none. The authors used whole genome sequencing of branch tips of a Eucalyptus tree previously phenotypically shown to have at least one somatic mutation affecting herbivore resistance. The novelty of this paper is an effective phylogenetic approach to identifying somatic mutations across a single tree, adjusting the estimated number of mutations through in silico estimation of false positive rates (which were low) and false negative rates (which were quite high), and putting the estimated mutation rate into context with comparisons with Arabidopsis and other plants including trees. The phylogenetic topology based on somatic mutations largely matched the topology of the actual tree. The authors provide further support that long-lived and large plants seem to have apical meristems that reduce the likelihood of accumulating somatic mutations compared to short-lived, small statured species, a hypothesis the senior author of this paper has previously written eloquently about.

My comments are relatively minor - mainly asking for some clarifications in the event they will add to the interpretation of the results.

l. 59-60 Branch tips were selected to maximize branch length distance between samples. This is nicely illustrated in Figure 1. Was any consideration given to the order of the branches sampled, i.e., how many new lateral meristems had been initiated along the branch path?

l. 123 - Were the 90 high-confidence somatic mutations identified all cases where a homozygous site became a heterozygous site?

l. 127-128 - why were false negative rates so high? It seems this should be discussed. The false negative approach is very low, and provides good evidence of the strength of the sampling approach used (with three technical replicates per branch tip).

l. 141 - This is a wide age range estimate. What is it based on? Does this species produce annual rings, i.e., is it possible to age the tree?

l. 161-164 - "While this per-generation rate is high, the rate per meter of growth is very low, roughly an order of magnitude lower than our estimate for *E. melliodora*, at around 3.5×10^{-10} (2.7×10^{-8} divided by the average height of individual Sitka spruce studied of 76 m), which is strikingly similar to the rate that we estimated here." At first read, this sentence appears to be contradictory. I think you mean the rate per generation in Sitka spruce was similar, but the rate per m of growth was about ten times lower?

l. 169-180 - I would have liked to see some discussion of why age vs size might matter to somatic mutation rates?

l. 249-256 and 289-298 - Which of these filtering steps that impressively reduced the false positive rate likely had the greatest impact on the false negative rate?

l. 340-345 - This should be in the results, not in the methods.

Review form: Reviewer 3

Recommendation

Accept with minor revision (please list in comments)

Scientific importance: Is the manuscript an original and important contribution to its field?

Good

General interest: Is the paper of sufficient general interest?

Excellent

Quality of the paper: Is the overall quality of the paper suitable?

Good

Is the length of the paper justified?

Yes

Should the paper be seen by a specialist statistical reviewer?

No

Do you have any concerns about statistical analyses in this paper? If so, please specify them explicitly in your report.

No

It is a condition of publication that authors make their supporting data, code and materials available - either as supplementary material or hosted in an external repository. Please rate, if applicable, the supporting data on the following criteria.

Is it accessible?

Yes

Is it clear?

Yes

Is it adequate?

Yes

Do you have any ethical concerns with this paper?

No

Comments to the Author

In "A phylogenomic approach reveals a low somatic mutation rate in a long-lived plantThe analysis, although interesting is relatively superficial." Orr and Colleagues present a simple and elegant study of the patterns of mutation across a single tree. The analysis is conducted assuming that somatic mutations from different parts of the tree will reflect the branching patterns in the tree itself. The analyses presented are quite simple but demonstrate that there is a relatively low rate of somatic mutation given that the tree is large and long lived. The paper also fits into an emerging literature exploring the long-held assumption that plants would have high mutation rates because they don't have an animal-like germline. Overall, I think the paper is clear and well written. I think the data are appropriate and the paper addresses a question that should be broadly interesting. I have a few comments that I think should be addressed before publication that I outline below

Format: Is this short manuscript style acceptable for Proc B? The paper is so short that many details are left to the methods or perhaps not provided at all. I think a bit more detail about how rates are calculated would be helpful.

Mutation results from *A. thaliana* (doi: 10.1016/j.cub.2016.03.067) has indicated that the mutation count is correlated not with branch length but by the number of branchings in which the meristem is forced to divide. Is there any evidence of this finding here?

The authors make some interesting conjecture about selection on somatic mutation - Perhaps the patterns of synonymous v non-synonymous mutation could be presented to quantify this claim.

Are mutation rates in the 'phylogeny' linear? - ie is the rate between two proximal tips proportional to the rate between distant tips?

L54: "Third, the approach allows us to estimate the false negative and the false positive rate of our inferences directly from the replicate samples (see below, and methods)" # 3 seems redundant with #1 - if its not redundant it should be more clearly explained.

L97: "significant correlation between the branch lengths of the physical tree and the maximum-parsimony tree of the same topology" I would be interested to see this plot. In other studies (oak) I thought they found no evidence for a correlation of branch length and mutation. Perhaps this difference could be discussed

L118: "We next developed a full maximum likelihood framework to detect somatic mutations" - I think it would be good to cite DeNovoGear here - the way it is written I was expecting to see a completely novel model based method to call mutations - perhaps I have misunderstood, but the method used is repurposing an already published ML analysis.

L124: "We estimated the false negative rate by creating 14,000 in silico somatic mutations in the raw reads²³, comprised of 1000 in silico mutations for each of the 14 branches of the physical tree, and measuring the recovery rate of these in silico mutations using our maximum likelihood approach. We were able to recover 4193 of the in silico mutations, suggesting that our recovery rate is 29.95%, and thus our false negative rate is 70.05%."

* A false negative is only a problem when you include sites like it in the denominator of rate calculations - what this means to me is that you can only view 30% of the genome- I assume that this accounted for in rate calculations

* Was the number of missed mutations even across branches? I assume quality and coverage vary by sample but also the quality of the genome might systematically obscure mutations across all samples.

How are replicates treated in the analysis - if you call each replicate genotype separately it may be harder to make high quality heterozygous calls with depth $\sim 10x$ /replicate. However, if the replicates are combined the depth increases to 30X and may allow more mutations to be identified. Which method was used?

If you found 90 mutations what are the "99" high confidence mutations used to build the tree. Were some of these extra 9 false positives and could they be the reason that there are multiple most parsimonious trees?

Is cell density similar across tissues? Presumably the rate should be correlated with cell number not size. I know it's not likely possible to analyze this explicitly but perhaps it could be discussed

149 - The number of mutations inherited by any one seed would not be the sum of the mutations across all branches. This means that each branch more or less should have their own mutations. The calculation of per generation mutation rate seems complex. The authors mention "with some addition assumptions" - can they expand on these assumptions and how on this per generation calculation is arrived at?

L162: This sentence is quite confusing and the end seems to contradict the beginning: "roughly an order of magnitude lower than our estimate for *E. melliodora*, at around 3.5×10^{-10} (2.7×10^{-8} divided by the average height of individual Sitka spruce studied of 76 m) which is strikingly similar to the rate that we estimate here

L163: How much does the height of a tree matter - isn't it the branching lengths that matter?

L177: low rates of somatic mutation in large plants - are the rates lower than in small herbaceous or annual plants?

Methods

L246: "9,679,544 potential variable sites" - this is a lot given you were removing these variants with the iterative mapping method

L251: "the depth is less than or equal to 500, since excessive depth is a signal of alignment issues" Is this total depth? Ie 500/24 ~20x depth for a sample? Normally people use something like 2x each sample's mean coverage.

252 "(v) the ExcessHet annotation was less than or equal to 40, since excessive heterozygosity at a site is a sign of genotyping errors" - This doesn't make sense because data were from one tree and a heterozygous site in the plant's genome should be heterozygous throughout the tree unless gene conversion or mutation changes it. Therefore all heterozygous sites will look to be in excess. You should check how big of a problem this is (ie how many mutations did it exclude)

Decision letter (RSPB-2019-2364.R0)

04-Dec-2019

Dear Dr Lanfear:

Your manuscript has now been peer reviewed and the reviews have been assessed by an Associate Editor. The reviewers' comments (not including confidential comments to the Editor) and the comments from the Associate Editor are included at the end of this email for your reference. As you will see, the reviewers and the Editors have raised some concerns with your manuscript and we would like to invite you to revise your manuscript to address them. However, the concerns are fairly modest, yet numerous.

Research ethics:

Use of animals and field studies:

Please submit a copy of your revised paper within three weeks. If we do not hear from you within this time your manuscript will be rejected. If you are unable to meet this deadline please let us know as soon as possible, as we may be able to grant a short extension.

Best wishes,

Professor John Hutchinson, Editor

Associate Editor

Comments to Author:

The manuscript by Orr et al. has now been seen by three external referees whose comments appear below. I have re-read the manuscript in light of their remarks and agree that this is indeed a clearly-written and compelling study.

On the other hand, all referees have made quite extensive suggestions for clarifications and improvements, and some of their suggestions overlap and can thus be met with a 'pooled' response by the authors (e.g., concerns about the reasons for the high false negative rate, questions about the handling of data from the three replicates per branch tip, concerns about the issue of (physical) tree branch length vs. the actual branching pattern (SAM to axillary meristems issue), and the mutation rate comparison to published data on other plants).

The authors should take all points raised by the referees into account in revising their manuscript. In addition, it is clear that the submitted version is not formatted in the style of Proc B; it needs to have clearly labeled sections such as Introduction, Methods, Results, Discussion (or Results and Discussion), and Acknowledgements. As is, I find the M&M section to be overly long in comparison to the main text, and this can/should be modified to some extent, as e.g. commented on by referee 3 (more details and discussion in main text).

I look forward to see a revised version of this manuscript.

Sincerely yours,
Thomas Städler, Ph.D.

Reviewer(s)' Comments to Author:

Referee: 1

Comments to the Author(s)

I had read this paper in preprint arxiv and in fact I like this paper. So I'm delighted to have the chance to give a formal review on it. The somatic mutations in large long-lived trees are attracting increasing interests from the community, and the current work by the authors could certainly serve as an excellent paradigm to develop related methodologies. Though I would like to see a quick publication of this work, I may still have to be rigorous to ensure this paper to be published as qualified as it could be.

My general concern is about the many assumptions made in this paper, some of which seem to be weak and could limit the application of the proposed method. The details are given below.

The phylogenomic approach is indeed intriguing, but related tests depend on the known physical structure of the tree. The clear ontogeny is not always easy to obtain on many early-branching or multi-stemmed trees. Besides, old large trees are vulnerable to environmental stresses, and old branches could be damaged while new branches could be regenerated near wounded sites. These factors complicate the true ontology as well as the underlying relationship to phylogeny. It's thus not clear whether the phylogenomic approach is robust enough against all these confounding factors.

Page 3 Line 46. Using three biological replicates could certainly reduce many non-context-dependent errors (i.e., errors assumed to be random by position), but not always true for context-dependent errors. Could we check how many false-positives are eliminated by checking if we only require one or two replicates in mutation calling. In turn, is it possible that this may also lead to false negatives as any replicate with poor coverage on certain sites could cause a rejection of true mutations.

Page 7 Line 127. What causes this very high false negative rate here? My guess is that a high divergence with reference genome could lead to fewer regions accessible, and removing repetitive regions also reduce the accessible regions. How many regions are actually valid in variant calling and how many regions are repetitive defined by RepeatMasker then? Or are there other factors lead to this high false negative rate?

Page 7 Line 128. Using simulation to estimate the false positives is nearer to a debugging process, which means most times if we find these false positives we could re-adjust the criteria to eliminate them. If we could not in fact reject them, even after manual inspection, how can we be sure that they are truly false positives? So how do the 11 false-positive mutations look like? Are they failed any criteria or just not match well with the ontology?

Page 7 Line 136. By calculating per meter mutation rate, the implied assumption here is that mutations fixed linearly with the branch length. I had to say that I don't agree very well on this assumption. There are growing bodies of evidences suggest that the branching processes (e.g., from SAM to axillary meristems) rather than the elongation processes generating most fixed mutations. Considering two branch tips with same lengths to the ground but with different branching orders, the higher-ordered branch tip could have higher per meter mutation rate as it undergoes much more stem cell divisions.

Referee: 2

Comments to the Author(s)

This is a very interesting, clear, well-written paper on a current hot topic - that of the extent of somatic mutations in plants in general, and trees in particular. The potential contribution of somatic mutations to genetic diversity of long-lived plants has been a topic of theoretical discussion for decades, but only recently has sequencing technology and cost been sufficient to address it empirically. There have been several papers on this topic in the past ~2 years - prior to that, there were none. The authors used whole genome sequencing of branch tips of a Eucalyptus tree previously phenotypically shown to have at least one somatic mutation affecting herbivore resistance. The novelty of this paper is an effective phylogenetic approach to identifying somatic mutations across a single tree, adjusting the estimated number of mutations through in silico estimation of false positive rates (which were low) and false negative rates (which were quite high), and putting the estimated mutation rate into context with comparisons with Arabidopsis and other plants including trees. The phylogenetic topology based on somatic mutations largely matched the topology of the actual tree. The authors provide further support that long-lived and large plants seem to have apical meristems that reduce the likelihood of accumulating somatic mutations compared to short-lived, small statured species, a hypothesis the senior author of this paper has previously written eloquently about.

My comments are relatively minor - mainly asking for some clarifications in the event they will add to the interpretation of the results.

l. 59-60 Branch tips were selected to maximize branch length distance between samples. This is nicely illustrated in Figure 1. Was any consideration given to the order of the branches sampled, i.e., how many new lateral meristems had been initiated along the branch path?

l. 123 - Were the 90 high-confidence somatic mutations identified all cases where a homozygous site became a heterozygous site?

l. 127-128 - why were false negative rates so high? It seems this should be discussed. The false negative approach is very low, and provides good evidence of the strength of the sampling approach used (with three technical replicates per branch tip).

l. 141 - This is a wide age range estimate. What is it based on? Does this species produce annual rings, i.e., is it possible to age the tree?

l. 161-164 - "While this per-generation rate is high, the rate per meter of growth is very low, roughly an order of magnitude lower than our estimate for *E. melliodora*, at around 3.5×10^{-10} (2.7×10^{-8} divided by the average height of individual Sitka spruce studied of 76 m), which is strikingly similar to the rate that we estimated here." At first read, this sentence appears to be contradictory. I think you mean the rate per generation in Sitka spruce was similar, but the rate per m of growth was about ten times lower?

l. 169-180 - I would have liked to see some discussion of why age vs size might matter to somatic mutation rates?

l. 249-256 and 289-298 - Which of these filtering steps that impressively reduced the false positive rate likely had the greatest impact on the false negative rate?

l. 340-345 - This should be in the results, not in the methods.

Referee: 3

Comments to the Author(s)

In "A phylogenomic approach reveals a low somatic mutation rate in a long-lived plant The analysis, although interesting is relatively superficial." Orr and Colleagues present a simple and elegant study of the patterns of mutation across a single tree. The analysis is conducted assuming that somatic mutations from different parts of the tree will reflect the branching patterns in the tree itself. The analyses presented are quite simple but demonstrate that there is a relatively low rate of somatic mutation given that the tree is large and long lived. The paper also fits into an emerging literature exploring the long-held assumption that plants would have high mutation rates because they don't have an animal-like germline. Overall, I think the paper is clear and well written. I think the data are appropriate and the paper addresses a question that should be broadly interesting. I have a few comments that I think should be addressed before publication that I outline below

Format: Is this short manuscript style acceptable for Proc B? The paper is so short that many details are left to the methods or perhaps not provided at all. I think a bit more detail about how rates are calculated would be helpful.

Mutation results from *A. thaliana* (doi: 10.1016/j.cub.2016.03.067) has indicated that the mutation count is correlated not with branch length but by the number of branchings in which the meristem is forced to divide. Is there any evidence of this finding here?

The authors make some interesting conjecture about selection on somatic mutation - Perhaps the patterns of synonymous v non-synonymous mutation could be presented to quantify this claim.

Are mutation rates in the 'phylogeny' linear? - ie is the rate between two proximal tips proportional to the rate between distant tips?

L54: "Third, the approach allows us to estimate the false negative and the false positive rate of our inferences directly from the replicate samples (see below, and methods)" # 3 seems redundant with #1 - if its not redundant it should be more clearly explained.

L97: "significant correlation between the branch lengths of the physical tree and the maximum-parsimony tree of the same topology" I would be interested to see this plot. In other studies (oak) I thought they found no evidence for a correlation of branch length and mutation. Perhaps this difference could be discussed

L118: "We next developed a full maximum likelihood framework to detect somatic mutations" - I think it would be good to cite DeNovoGear here - the way it is written I was expecting to see a

completely novel model based method to call mutations - perhaps I have misunderstood, but the method used is repurposing an already published ML analysis.

L124: "We estimated the false negative rate by creating 14,000 in silico somatic mutations in the raw reads²³, comprised of 1000 in silico mutations for each of the 14 branches of the physical tree, and measuring the recovery rate of these in silico mutations using our maximum likelihood approach. We were able to recover 4193 of the in silico mutations, suggesting that our recovery rate is 29.95%, and thus our false negative rate is 70.05%."

* A false negative is only a problem when you include sites like it in the denominator of rate calculations - what this means to me is that you can only view 30% of the genome- I assume that this accounted for in rate calculations

* Was the number of missed mutations even across branches? I assume quality and coverage vary by sample but also the quality of the genome might systematically obscure mutations across all samples.

How are replicates treated in the analysis - if you call each replicate genotype separately it may be harder to make high quality heterozygous calls with depth $\sim 10x$ /replicate. However, if the replicates are combined the depth increases to 30X and may allow more mutations to be identified. Which method was used?

If you found 90 mutations what are the "99" high confidence mutations used to build the tree. Were some of these extra 9 false positives and could they be the reason that there are multiple most parsimonious trees?

Is cell density similar across tissues? Presumably the rate should be correlated with cell number not size. I know it's not likely possible to analyze this explicitly but perhaps it could be discussed

149 - The number of mutations inherited by any one seed would not be the sum of the mutations across all branches. This means that each branch more or less should have their own mutations. The calculation of per generation mutation rate seems complex. The authors mention "with some addition assumptions" - can they expand on these assumptions and how on this per generation calculation is arrived at?

L162: This sentence is quite confusing and the end seems to contradict the beginning: "roughly an order of magnitude lower than our estimate for *E. melliodora*, at around 3.5×10^{-10} (2.7×10^{-8} divided by the average height of individual Sitka spruce studied of 76 m) which is strikingly similar to the rate that we estimate here

L163: How much does the height of a tree matter - isn't it the branching lengths that matter?

L177: low rates of somatic mutation in large plants - are the rates lower than in small herbaceous or annual plants?

Methods

L246: "9,679,544 potential variable sites" - this is a lot given you were removing these variants with the iterative mapping method

L251: "the depth is less than or equal to 500, since excessive depth is a signal of alignment issues" Is this total depth? Ie 500/24 $\sim 20x$ depth for a sample? Normally people use something like 2x each sample's mean coverage.

252 "(v) the ExcessHet annotation was less than or (v) equal to 40, since excessive heterozygosity at a site is a sign of genotyping errors" - This doesn't make sense because data were from one tree and a heterozygous site in the plant's genome should be heterozygous throughout the tree unless gene conversion or mutation changes it. Therefore all heterozygous sites will look to be in excess. You should check how big of a problem this is (ie how many mutations did it exclude)

Author's Response to Decision Letter for (RSPB-2019-2364.R0)

See Appendix A.

Decision letter (RSPB-2019-2364.R1)

11-Feb-2020

Dear Dr Lanfear

I am pleased to inform you that your manuscript RSPB-2019-2364.R1 entitled "A phylogenomic approach reveals a low somatic mutation rate in a long-lived plant" has been accepted for publication in Proceedings B. Congratulations!!

The referee(s) have recommended publication, but also suggest some minor revisions to your manuscript. Therefore, I invite you to respond to the referee(s)' comments and revise your manuscript. Because the schedule for publication is very tight, it is a condition of publication that you submit the revised version of your manuscript within 7 days. If you do not think you will be able to meet this date please let us know.

Sincerely,

Professor John Hutchinson

Associate Editor:

Board Member

Comments to Author:

The manuscript by Orr et al. is a thoroughly revised version of their original submission, which was scrutinised extensively by three referees. The authors have provided a very thorough point-by-point list of their responses to all reviewers' remarks and the resulting changes to the MS. I am convinced that this revised version does justice to the reviewers' concerns, and that it represents a markedly improved account and documentation of their study.

I suggest a few minor changes to bring the MS in line with the journal's standards. As is, the M&M section is still at the end of the main text, but it should precede "Results and Discussion". This change will affect the order of citations: currently, citations 37-44 are first cited in the M&M section, and 1-20 cited in "Background" (Introduction). The citation style in the text is still in "Nature" format, as may be the list of References - in any case, they are not in Proceedings B format.

I also suggest to substructure the long "Results and Discussion" section, as is done for the M&M section, and to number MS sections sequentially. With ca. 120 words, the Abstract seems short; there should be space to provide more details and/or rationale if desired.

A few apparent typos:

- * line 129: should read "means" instead of "mean".
- * line 193: it should read "...estimate the per-generation..." (i.e., superfluous "that").
- * line 322: the citation to the Phangorn R package should be (currently) no. 41, not 36?
- * line 357: "...had a log10 likelihood...".
- * lines 385-386: it should read "...recorded how many of the ..", and "...false negative rate is..".
- * 4 of the listed References seem to have incomplete information: no. 7 (Klekowski & Godfrey 1989) has wrong volume number. No. 8 (Ally et al. 2010) lacks an article ID or page numbers. No. 24 (Kainer et al. 2019) should have a DOI (if not volume/page numbers). No. 33 (Scofield 2014) lacks name of the journal.

Yours sincerely,
Thomas Städler, Ph.D.

Decision letter (RSPB-2019-2364.R2)

18-Feb-2020

Dear Dr Lanfear

I am pleased to inform you that your manuscript entitled "A phylogenomic approach reveals a low somatic mutation rate in a long-lived plant" has been accepted for publication in Proceedings B.

Your article has been estimated as being 9 pages long. Our Production Office will be able to confirm the exact length at proof stage.

Open Access

Paper charges

Sincerely,

Proceedings B

Appendix A

04-Dec-2019

Dear Dr Lanfear:

Your manuscript has now been peer reviewed and the reviews have been assessed by an Associate Editor. The reviewers' comments (not including confidential comments to the Editor) and the comments from the Associate Editor are included at the end of this email for your reference. As you will see, the reviewers and the Editors have raised some concerns with your manuscript and we would like to invite you to revise your manuscript to address them. However, the concerns are fairly modest, yet numerous.

Research ethics:

Use of animals and field studies:

If your study uses animals please include details in the methods section of any approval and licences given to carry out the study and include full details of how animal welfare standards were ensured. Field studies should be conducted in accordance with local legislation; please

include details of the appropriate permission and licences that you obtained to carry out the field work.

If you wish to submit your data to Dryad (<http://datadryad.org/>) and have not already done so you can submit your data via this link [http://datadryad.org/submit?journalID=RSPB&manu=\(Document](http://datadryad.org/submit?journalID=RSPB&manu=(Document) not available), which will take you to your unique entry in the Dryad repository.

Please submit a copy of your revised paper within three weeks. If we do not hear from you within this time your manuscript will be rejected. If you are unable to meet this deadline please let us know as soon as possible, as we may be able to grant a short extension.

Best wishes,

Professor John Hutchinson, Editor
mailto: proceedingsb@royalsociety.org

Associate Editor

Comments to Author:

The manuscript by Orr et al. has now been seen by three external referees whose comments appear below. I have re-read the manuscript in light of their remarks and agree that this is indeed a clearly-written and compelling study.

On the other hand, all referees have made quite extensive suggestions for clarifications and improvements, and some of their suggestions overlap and can thus be met with a 'pooled' response by the authors (e.g., concerns about the reasons for the high false negative rate, questions about the handling of data from the three replicates per branch tip, concerns about the issue of (physical) tree branch length vs. the actual branching pattern (SAM to axillary meristems issue), and the mutation rate comparison to published data on other plants).

The authors should take all points raised by the referees into account in revising their manuscript. In addition, it is clear that the submitted version is not formatted in the style of Proc B; it needs to have clearly labeled sections such as Introduction, Methods, Results, Discussion (or Results and Discussion), and Acknowledgements. As is, I find the M&M section to be overly long in comparison to the main text, and this can/should be modified to some extent, as e.g. commented on by referee 3 (more details and discussion in main text).

Response: we have responded to all of the points raised by the reviewers below, but wanted to add here one thing which is not mentioned explicitly below: the relative lengths of the M&M versus the main text. In the original submission the M&M was ~51% of the paper (excluding abstract and references). In the revised document the M&M is ~38% of the paper. We have reduced the M&M as much as possible, while retaining sufficient detail on the approach that others could repeat it along with our open-source code and open-access data.

I look forward to see a revised version of this manuscript.

Sincerely yours,

Thomas Städler, Ph.D.

Reviewer(s)' Comments to Author:

Referee: 1

Comments to the Author(s)

I had read this paper in preprint arxiv and in fact I like this paper. So I'm delighted to have the chance to give a formal review on it. The somatic mutations in large long-lived trees are attracting increasing interests from the community, and the current work by the authors could certainly serve as an excellent paradigm to develop related methodologies. Though I would like to see a quick publication of this work, I may still have to be rigorous to ensure this paper to be published as qualified as it could be.

My general concern is about the many assumptions made in this paper, some of which seem to be weak and could limit the application of the proposed method. The details are given below.

The phylogenomic approach is indeed intriguing, but related tests depend on the known physical structure of the tree. The clear ontogeny is not always easy to obtain on many early-branching or multi-stemmed trees. Besides, old large trees are vulnerable to environmental stresses, and old branches could be damaged while new branches could be regenerated near wounded sites. These factors complicate the true ontology as well as the underlying relationship to phylogeny. It's thus not clear whether the phylogenomic approach is robust enough against all these confounding factors.

Response: we agree that the method has limitations, and have edited the manuscript in two places to more clearly represent them. Specifically, beginning on line 75 we now state:

“This approach relies on being able to infer the ontogeny of the tree with sufficient accuracy that a valid comparison can be made between the ontogeny of the tree and a phylogeny generated from that tree's somatic variants. Documenting a plant's ontogeny with sufficient accuracy may not be possible for all plant species or individuals. Nevertheless, the physical structure of the tree we studied was clear (Figure 1), and although Eucalyptus trees are known to frequently lose branches, branch loss and regrowth should not affect the correlation between ontogeny and phylogeny provided that sufficient mutations accumulate during cell replication.”

and we have added at the end of the same paragraph the observation that difficulty in inferring ontogeny could produce false negatives for the positive control, but is unlikely to produce false positives (starting on line 106):

“Notably, while various factors such as the difficulty of correctly inferring plant ontogeny may limit the utility of a phylogenetic positive control such as we present here (i.e. may produce false negative results in which the structure of the tree appears, erroneously, to differ from the phylogeny of the sequenced genomes), it is unlikely that these factors would erroneously cause a close match between the physical structure of the tree and a phylogeny generated from the

genomes of 8 branches of that tree (i.e. a false positive). We therefore conclude that these analyses demonstrate that the phylogeny recovered from the genomic data matches the physical structure of the tree, and confirm that there is strong biological signal in our data.”

Page 3 Line 46. Using three biological replicates could certainly reduce many non-context-dependent errors (i.e., errors assumed to be random by position), but not always true for context-dependent errors. Could we check how many false-positives are eliminated by checking if we only require one or two replicates in mutation calling. In turn, is it possible that this may also lead to false negatives as any replicate with poor coverage on certain sites could cause a rejection of true mutations.

Response: we have now repeated the entire analysis pipeline from the positive control analysis when requiring just one or two replicates in mutation calling. Full details of this are provided in section 1 of the supplementary material. To summarise those results - the outcome is as expected by the reviewer and ourselves: the false negative is highest when we require all three genotypes to agree (simply because we filter out the most sites in that analysis compared to the other two), and the number of false positive mutations increases dramatically when one reduces the stringency of the replicate filter. For example, simply shifting from a 3-replicates-must-agree filter to a 2-replicates-must-agree filter, we infer 99 mutations, with a false negative rate of 77% and a false discovery rate of 56% in the first case, and 335K mutations a false negative rate of 68% and a false discovery rate of 97.5% in the second case.

So, while it is certainly true that our filter leads to more false negatives, it is also true (and important to our analyses) that it leads to vastly fewer false positives. We describe this in the last paragraph of section 1 of the supplementary information as follows:

“For the positive control analyses in which we used the replicate filter, the numbers in table 1.1 show the importance of using three replicate filters. For example, our phylogenetic positive control would be extremely unlikely to be informative if we used an alignment of 335,867 mutations (as recovered when we require only two genotypes to agree) of which 327,351 are likely to be false positives, simply because the phylogenetic signal from the 2.5% of the data which are true mutations is likely to be swamped by the 97.5% of the data which are false positives. On the other hand, a dataset of 99 variable sites of which roughly 56 are false positives (and thus roughly 43 are likely to be true positives) has a much better chance of recovering the expected phylogenetic tree.”

We hope this provides some clarity on the effects of the replicate filter on the false negative rate and the false discovery rate.

We have also modified the main text to point out that these analyses were carried out. Specifically, we reference the supplementary information in the Results and Discussion when we describe the filters used in the positive control pipeline (starting on line 84): “We then called variants using GATK22 in all three replicates of all eight branch tips, and used a set of strict

filters (see methods and supplementary information) designed to avoid false positive mutations in order to arrive at an alignment of 99 high-confidence somatic variants.”

We also refer to the supplementary information in the methods when we describe the replicate filtering (starting on line 304): “(i) genotype calls were identical within all three replicates of each branch tip (see also supplementary information section 1);”

Page 7 Line 127. What causes this very high false negative rate here? My guess is that a high divergence with reference genome could lead to fewer regions accessible, and removing repetitive regions also reduce the accessible regions. How many regions are actually valid in variant calling and how many regions are repetitive defined by RepeatMasker then? Or are there other factors lead to this high false negative rate?

Response: We have clarified these points in the main manuscript in two places (details below). We provide a longer response here.

The high false negative rate is caused primarily by our very stringent filters of sites. This is expected simply because the harder we work to avoid false positives (which is our primary concern), the more collateral we are likely to have with false negatives.

The divergence from the reference genome may contribute to the false negative rate, but this should be ameliorated to some extent by the creation of a pseudo-reference genome, and the fact that most Eucalyptus genome are structurally very similar to one another.

In the DeNovoGear analysis for which we analyse the false negative rate in the main text, we do not exclude any of the genome for the analysis because our model should be able to infer mutations in some repeat regions (e.g. those that are small, appear on large haplotype blocks, and have no alignment issues flagged by our filters), and will filter out unreliable (i.e. false-positive) putative mutations in other repeat regions due to our filters on mutation proximity and the required minimum length of haplotype blocks. Long repeats will tend to have short haplotype blocks and potentially a number of putative mutations - in both cases this will lead us to exclude unreliable mutations from repeat regions, while keeping reliable mutations such as those inferred in short repeats. In other words, we think our mutation proximity filter and our haplotype block length filter provide a better approach to dealing with repeats than simply removing all repeat regions from our analysis.

To answer the specific question the reviewer asked: RepeatMasker estimates that 39.8% of the genome consists of repeats (this is roughly as expected given the more rigorous analysis in the Eucalyptus grandis genome paper: <https://www.nature.com/articles/nature13308>). In our simulated mutations, 5182 of the 14,000 simulated mutations (37%) fell in repeat regions. Our DeNovoGear analysis was able to correctly detect 13% of the mutations that were simulated in repeat regions, and 40% of the mutations that occurred outside repeat regions. This confirms that detecting mutations in repeat regions is indeed more difficult than detecting mutations outside repeat regions. It also quantifies the contribution of this difficulty to our false negative

rate: our false negative rate is ~87% in the ~40% of the genome that consists of repeats, and ~60% in the ~60% of the non-repeat genome.

We have updated the main manuscript to summarise these details. Beginning at line 153 the manuscript now reads:

“Our ability to recover mutations differs substantially between repeat regions and non-repeat regions: we recover 40% of the simulated mutations in non-repeat regions, but just 13% of the simulated mutations in repeat regions (which make up roughly 40% of the genome). This difference is explained primarily by the stringent filters we use, that lead us to screen out many putative somatic mutations in repeat regions.”

We have also clarified this in the description of the filters beginning on line 354: “(1) were on a haplotype block with a size less than 500 nucleotides (among other things, this filter will remove many putative variants that fall in long repeat regions); (2) were within 1000 nucleotides of another de novo variant (indicative of alignment issues such as might occur in repeats and other regions)”

Page 7 Line 128. Using simulation to estimate the false positives is nearer to a debugging process, which means most times if we find these false positives we could re-adjust the criteria to eliminate them. If we could not in fact reject them, even after manual inspection, how can we be sure that they are truly false positives? So how do the 11 false-positive mutations look like? Are they failed any criteria or just not match well with the ontology?

Response: the potential similarity of using simulations to estimate false positives and a debugging process is a misunderstanding which we have now clarified in the manuscript on line 401 as follows: “We calculated the false discovery rate only once, after the details of the pipeline were finalised, to avoid overfitting our pipeline to artefactually reduce the false discovery rate.”

During the development of the pipeline, we did (of course!) examine samples of mutation calls by eye and refine the pipeline in an attempt to avoid false positives. However, we only calculated the number of false positives (and the false discovery rate) using the simulation process we describe in the “Estimation of the false discovery rate” once the other analyses were complete and the pipeline finalised. The reason we did not iterate is because this would likely be overfitting our analysis decisions to our data to optimise the false positive rate. It would also invalidate our estimate of the false discovery rate.

The 11 false positive mutations are almost all cases in which the randomized tree strongly implies that a mutation occurred at a site where there is somewhat weak evidence for a particular genotype in all replicates. I.e. a site where there is little evidence for a mutation on the physical tree topology, but in which the randomisation of the topology induces a false positive mutation that happens to pass all of our filters. Given the stringent filtering, such cases are rare. (Though section 1 of the supplementary information highlights that much higher false positive rates are common with less sophisticated ways of using the replicate information, e.g. if rather

than using the DeNovoGear model we simply require that at least 2 of the three replicate genotypes agree, we estimate that more than 97% of the inferred mutations are false positives).

Page 7 Line 136. By calculating per meter mutation rate, the implied assumption here is that mutations fixed linearly with the branch length. I had to say that I don't agree very well on this assumption. There are growing bodies of evidences suggest that the branching processes (e.g., from SAM to axillary meristems) rather than the elongation processes generating most fixed mutations. Considering two branch tips with same lengths to the ground but with different branching orders, the higher-ordered branch tip could have higher per meter mutation rate as it undergoes much more stem cell divisions.

Response: thanks for pointing this out. We agree that evidence is mounting that mutations occur primarily at branching events rather than during elongation, and have now sought to both make this clear in the manuscript. Specifically, starting on line 177 (at the end of the paragraph in which we calculate the per-metre mutation rate) the manuscript now reads:

“It is important to note that it remains unclear whether mutations in growing plants accumulate linearly with the amount of physical growth. Indeed, evidence is accumulating that in at least some (and perhaps most) species, mutations may accumulate primarily at branching events rather than during elongation of individual branches^{26,27}. If this is the case, then the correlation we observe between the physical branch length and the number of inferred somatic mutations (see above, and supplementary information section 4) may be due to a correlation between the physical length of a branch and the number of branching events that occurred along that branch during the plant's development. It is not possible to directly estimate the number of branching events along each branch in the individual tree we used in this study, because we expect that the tree will have regularly lost branches throughout its life, leaving no accurate record of the number of branching events.”

Referee: 2

Comments to the Author(s)

This is a very interesting, clear, well-written paper on a current hot topic - that of the extent of somatic mutations in plants in general, and trees in particular. The potential contribution of somatic mutations to genetic diversity of long-lived plants has been a topic of theoretical discussion for decades, but only recently has sequencing technology and cost been sufficient to address it empirically. There have been several papers on this topic in the past ~2 years - prior to that, there were none. The authors used whole genome sequencing of branch tips of a Eucalyptus tree previously phenotypically shown to have at least one somatic mutation affecting herbivore resistance. The novelty of this paper is an effective phylogenetic approach to identifying somatic mutations across a single tree, adjusting the estimated number of mutations through in silico estimation of false positive rates (which were low) and false negative rates (which were quite high), and putting the estimated mutation rate into context with comparisons with Arabidopsis and other plants including trees. The phylogenetic topology based on somatic mutations largely matched the topology of the actual tree. The authors provide further support

that long-lived and large plants seem to have apical meristems that reduce the likelihood of accumulating somatic mutations compared to short-lived, small statured species, a hypothesis the senior author of this paper has previously written eloquently about.

My comments are relatively minor - mainly asking for some clarifications in the event they will add to the interpretation of the results.

I. 59-60 Branch tips were selected to maximize branch length distance between samples. This is nicely illustrated in Figure 1. Was any consideration given to the order of the branches sampled, i.e., how many new lateral meristems had been initiated along the branch path?

Response: unfortunately we did not take account of the branching order when we took the samples, as we took the samples years before the research suggesting branching order might be important was published. Nevertheless, we did revisit a set of detailed photographs of the fieldwork, and this led us to realise two things. First, that the photographs are not of sufficient detail to estimate the branching order. Second, that even if they were, branching order estimates from a relatively old tree that has likely lost a lot of branches may not be very accurate. Regardless, we appreciate that consideration of branching order now seems to be important, and we have edited the manuscript to reflect this. Specifically, starting on line 177 (at the end of the paragraph in which we calculate the per-metre mutation rate) the manuscript now reads:

“It is important to note that it remains unclear whether mutations in growing plants accumulate linearly with the amount of physical growth. Indeed, evidence is accumulating that in at least some (and perhaps most) species, mutations may accumulate primarily at branching events rather than during elongation of individual branches^{26,27}. If this is the case, then the correlation we observe between the physical branch length and the number of inferred somatic mutations (see above, and supplementary information section 4) may be due to a correlation between the physical length of a branch and the number of branching events that occurred along that branch during the plant’s development. It is not possible to directly estimate the number of branching events along each branch in the individual tree we used in this study, because we expect that the tree will have regularly lost branches throughout its life, leaving no accurate record of the number of branching events.”

I. 123 - Were the 90 high-confidence somatic mutations identified all cases where a homozygous site became a heterozygous site?

Response: No. 69 calls were homozygous to heterozygous, and 21 calls were heterozygous to homozygous.

I. 127-128 - why were false negative rates so high? It seems this should be discussed. The false negative approach is very low, and provides good evidence of the strength of the sampling approach used (with three technical replicates per branch tip).

Response: the short answer to this is that the false negative rates were high because we used a very strict set of filters. We have now discussed this more in a number of places throughout the manuscript. The main manuscript starting on line 118 now reads:

“This method improves on the approach we used in our positive control, above, because it increases our power to detect true somatic mutations and avoid false positives by assuming that the phylogenetic structure of the samples follows the physical structure of the tree, an assumption that is validated by the analyses above. It also makes better use of the replicate sampling design than the method we use for our positive control, above, by directly modelling the expected variation in sequencing data across our three biological replicates under the expectation that all three replicates were sequenced from a single underlying genotype (see methods and supplementary information).”

And starting on line 153 we now also discuss in more detail the false negative rate in repeat regions versus non-repeat regions:

“Our ability to recover mutations differs substantially between repeat regions and non-repeat regions: we recover 40% of the simulated mutations in non-repeat regions, but just 13% of the simulated mutations in repeat regions (which make up roughly 40% of the genome). This difference is explained primarily by the stringent filters we use, that lead us to screen out many putative somatic mutations in repeat regions.”

The supplementary information also contains a suite of re-analyses that provide more detail on the false negative and false positive mutations generated by our analyses. These include: the positive control analyses for which we discuss in detail the effect of the replicate filters on the false negative and false discovery rate (supp. Info section 1); an analysis of the false negative rate among different branches of the tree (supp. Info section 2); the effects of the ExcessHet filter on false negative and false discovery rates (supp. Info section 3); the effects of different filtering steps on the false negative rate and the number of false positive mutations (supp. Info section 5).

We hope that these additions provide a lot more insight into how we were able to reduce the false discovery rate, why we did so, and why this also led to the false negative rate being high.

I. 141 - This is a wide age range estimate. What is it based on? Does this species produce annual rings, i.e., is it possible to age the tree?

Response: unfortunately it is not possible to age the tree accurately. The tree grows in a temperate climate, and does not have growth rings (the only eucalypts with growth rings are the snowgums, which grow at high enough altitude that there is sufficient annual variation in their

growth rates to form rings). One can try to estimate its age from its diameter at breast height, but the formulas for doing so were developed for trees in different climates and we considered that using such an approach would most likely give an unwarranted sense of precision. We therefore chose instead to simply specify a wide range of ages that almost certainly contains the true age. The minimum bound of 50 years is reasonable as we have pictures of the tree from the 1980s where it looks very similar to today. It thus seems unlikely that the tree could be younger than 50 years. The upper bound of 200 years reflects the fact that most eucalypts are relatively short lived (<200 years) and this tree is certainly not of a size that would suggest it is very old. We have now clarified this in the manuscript such that the relevant sentence now reads (starting on line 172):

“Although the exact age of this individual is unknown and difficult to estimate – it lives in a temperate climate and does not produce growth rings – its age is nevertheless almost certainly between 50 and 200 years old.”

I. 161-164 - "While this per-generation rate is high, the rate per meter of growth is very low, roughly an order of magnitude lower than our estimate for *E. melliodora*, at around 3.5×10^{-10} (2.7×10^{-8} divided by the average height of individual Sitka spruce studied of 76 m), which is strikingly similar to the rate that we estimated here." At first read, this sentence appears to be contradictory. I think you mean the rate per generation in Sitka spruce was similar, but the rate per m of growth was about ten times lower?

Response: we thank the reviewer for highlighting this remarkably unclear sentence. We have now edited it to correct this, and also to clarify why we calculated the per m mutation rate for Sitka spruce as we did (starting line 209):

“While this per-generation rate is very similar to the one we estimate here, the rate of somatic mutation per meter of growth is around an order of magnitude lower in the Sitka spruce than our estimate for *E. melliodora* (2.75×10^{-9} somatic mutations per base pair per meter of growth for *E. melliodora* estimated here, versus 3.5×10^{-10} somatic mutations per base pair per meter of growth for Sitka spruce, estimated by dividing the per-generation mutation rate of 2.7×10^{-8} mutations per base by the average height of studied individuals of 76 m¹⁵, an appropriate calculation because the somatic mutation rate was estimated from paired samples taken from the base and the top of a collection of individual trees).”

I. 169-180 - I would have liked to see some discussion of why age vs size might matter to somatic mutation rates?

Response: thanks for the suggestion. We have now included a discussion of this point in the final paragraph of the discussion, which reads (starting line 231):

“Longevity and size are two aspects of plant life history likely to be of central importance to the evolution of somatic mutation rates. Larger plants may suffer from a higher accumulation of somatic mutations because of the necessity for additional cell divisions. Plants that live longer

may suffer from a higher accumulation of somatic mutations because of the accumulation of DNA damage over time, and/or increased cell turnover in long-lived tissues. The relative importance of these two factors may differ among clades, species, and individual tissues, and is likely to also depend on the balance between DNA damage and repair between cell divisions³⁶, the accuracy of DNA replication, cell size, and the rate of cell division.”

I. 249-256 and 289-298 - Which of these filtering steps that impressively reduced the false positive rate likely had the greatest impact on the false negative rate?

Response: we have now added a section to the supplementary information that describes for the positive control analysis exactly how many mutations were inferred at each step, and thus how many mutations were removed by each filter, as well as the false negative rate of the pipeline after the application of each filter, and the number of false positive mutations inferred. Examination of this table suggests that the three most important filters were likely the filtering out all variants near indels, the removal of variants inside repeat regions, and the requirement that all three replicate genotypes agree. This is perhaps not surprising, since the first two steps remove sizeable fractions of the genome from consideration (e.g. ~40% of the genome is removed when we filter out repeat regions), and the third step we now know (see supplementary information section 1) had a very large effect on reducing the FDR and concomitantly increasing the false negative rate.

I. 340-345 - This should be in the results, not in the methods.

Response: We have moved this section to the Results and Discussion section.

Referee: 3

Comments to the Author(s)

In "A phylogenomic approach reveals a low somatic mutation rate in a long-lived plant The analysis, although interesting is relatively superficial." Orr and Colleagues present a simple and elegant study of the patterns of mutation across a single tree. The analysis is conducted assuming that somatic mutations from different parts of the tree will reflect the branching patterns in the tree itself. The analyses presented are quite simple but demonstrate that there is a relatively low rate of somatic mutation given that the tree is large and long lived. The paper also fits into an emerging literature exploring the long-held assumption that plants would have high mutation rates because they don't have an animal-like germline. Overall, I think the paper is clear and well written. I think the data are appropriate and the paper addresses a question that should be broadly interesting. I have a few comments that I think should be addressed before publication that I outline below

Format: Is this short manuscript style acceptable for Proc B? The paper is so short that many details are left to the methods or perhaps not provided at all. I think a bit more detail about how rates are calculated would be helpful.

Response: we have added more detail throughout the paper, and rewritten the paper into sections that match the usual formatting for Proceedings B. For example, the length of the main text (excluding methods, but including the background, results, and discussion, and figure legends) has gone from 2074 words in the original submission to 3272 in the revised submission (~50% longer). The vast majority of the additional material is devoted to giving increased detail on the methods and analyses.

We have provided all details for all of the analyses and interpretations that we include in the paper. The methods provide verbal descriptions of all of our analyses, and we have kept these as concise as possible while providing sufficient detail to explain what we did. In addition to the text of the main paper, we now include a further 2000 word supplementary information which describes a suite of additional analyses. All of the data and code necessary to replicate all analyses in the paper are also publicly available, with extensive documentation to ensure that others can repeat our analyses. We have tried to ensure that our analyses are genuinely reproducible by asking an independent party to reproduce the analyses using the public data and public code repositories that accompany the paper. They were able to successfully replicate our analyses.

Mutation results from *A. thaliana* (doi: 10.1016/j.cub.2016.03.067) has indicated that the mutation count is correlated not with branch length but by the number of branchings in which the meristem is forced to divide. Is there any evidence of this finding here?

Response: thanks for the suggestion. Referee two had a very similar suggestion, and we wholeheartedly agree that this is an important point. For simplicity, we simply copy our response to referee 2 here (the response is the first one to referee 2).

Unfortunately we did not take account of the branching order when we took the samples, as we took the samples years before the research suggesting branching order might be important was published. Nevertheless, we did revisit a set of detailed photographs of the fieldwork, and this led us to realise two things. First, that the photographs are not of sufficient detail to estimate the branching order. Second, that even if they were, branching order estimates from a relatively old tree that has likely lost a lot of branches may not be very accurate. Regardless, we appreciate that consideration of branching order now seems to be important, and we have edited the manuscript to reflect this. Specifically, starting on line 177 (at the end of the paragraph in which we calculate the per-metre mutation rate) the manuscript now reads:

“It is important to note that it remains unclear whether mutations in growing plants accumulate linearly with the amount of physical growth. Indeed, evidence is accumulating that in at least some (and perhaps most) species, mutations may accumulate primarily at branching events rather than during elongation of individual branches^{26,27}. If this is the case, then the correlation we observe between the physical branch length and the number of inferred somatic mutations (see above, and supplementary information section 4) may be due to a correlation between the physical length of a branch and the number of branching events that occurred along that branch during the plant’s development. It is not possible to directly estimate the number of branching

events along each branch in the individual tree we used in this study, because we expect that the tree will have regularly lost branches throughout its life, leaving no accurate record of the number of branching events.”

The authors make some interesting conjecture about selection on somatic mutation - Perhaps the patterns of synonymous v non-synonymous mutation could be presented to quantify this claim.

Response: We observed 5 non-synonymous mutations and one synonymous mutation. We have now edited the paper to include these numbers in the main text, and acknowledge that the dn/ds ratio would be interesting but that we can't reliably calculate it. Beginning on line 127 the paper now reads:

“Of the 90 high-confidence variants we identified, 20 were in genes. Of these, six were in coding regions, with five non-synonymous mutations and one synonymous mutation. The small sample size of synonymous and non-synonymous mutations mean that we cannot provide a meaningful estimate of the ratio of non-synonymous to synonymous somatic mutations, although such an estimate would help to understand the extent to which somatic mutations may be under selection.”

Are mutation rates in the 'phylogeny' linear? - ie is the rate between two proximal tips proportional to the rate between distant tips?

Response: to a certain extent, yes. More specifically, we reported in the original submission that a linear correlation between physical branch length and the number of somatic mutations on each branch was significant. We now provide more information on this correlation in section 4 of the supplementary material, including the relevant plots. To cut a long story short, the strength of the is significant in statistical terms, but there is a lot of unexplained variation (the R^2 value is ~80% if the plot is forced through the origin, and ~40% if it isn't).

L54: "Third, the approach allows us to estimate the false negative and the false positive rate of our inferences directly from the replicate samples (see below, and methods)" # 3 seems redundant with #1 - if its not redundant it should be more clearly explained.

Response: We may have misunderstood what the reviewer is asking here, and if so we apologise. Our interpretation is that the reviewer is asking whether calculating the number of false positives is redundant if one knows the false negative rate.

I think in part this was due to our incorrect terminology. Throughout the original submission we used "false positive rate" where we should have used "false discovery rate". This has now been fixed throughout.

To answer the question that we *think* the reviewer was asking - it is not redundant to estimate the number of false positive mutations in a sample given that one knows the number of false negatives. For example, we identify 90 high-confidence mutations. The false negative rate tells us what proportion of mutations we are likely to have missed given how strictly we filter the data. But it does not tell us how many of these 90 mutations are likely to be false positive inferences.

To clarify this in the manuscript, we have now considerably edited the paragraph of the results and discussion where this is presented (the paragraph starts on line 143). Of particular note we now include the following beginning on line 168:

“It is necessary to estimate both the number of false negative mutations and the number of false positive mutations in order to estimate a somatic mutation rate. The former allows one to correct for the number of somatic mutations which a pipeline has failed to detect, while the latter allows one to correct for the number of somatic mutations which a pipeline has erroneously inferred.”

L97: "significant correlation between the branch lengths of the physical tree and the maximum-parsimony tree of the same topology" I would be interested to see this plot. In other studies (oak) I thought they found no evidence for a correlation of branch length and mutation. Perhaps this difference could be discussed

Response: we now show this plot in section 4 of the supplementary information, as well as the details (and plots) of the regression line when the regression is forced through the origin (which is the correct approach since a non-existent branch cannot have somatic mutations) versus when it is not forced through the origin (necessary to test if the confidence intervals of the intercept include the origin, which they do). The regression is significant in both cases.

We do not discuss this further in the manuscript for the simple reason that this relationship was determined as part of our positive control. We expect that our inferred mutations in our positive control approach will be much less accurate than the set of 90 high-confidence variants inferred our maximum likelihood model in DeNovoGear. We cannot repeat the analysis with the mutations inferred with DeNovoGear because the structure of the tree is assumed as part of that analysis. Thus, we do not want to make too much of this result except to present it as a small part of the positive control.

L118: "We next developed a full maximum likelihood framework to detect somatic mutations" - I think it would be good to cite DeNovoGear here - the way it is written I was expecting to see a completely novel model based method to call mutations - perhaps I have misunderstood, but the method used is repurposing an already published ML analysis.

Response: we have edited the section as suggested, to read (starting on line 115): “We next developed a full maximum likelihood framework that extends the existing models in DeNovoGear (Ramu et al 2013) to detect somatic mutations in a phylogenetic context...”

L124: "We estimated the false negative rate by creating 14,000 in silico somatic mutations in the raw reads²³, comprised of 1000 in silico mutations for each of the 14 branches of the physical tree, and measuring the recovery rate of these in silico mutations using our maximum likelihood approach. We were able to recover 4193 of the in silico mutations, suggesting that our recovery rate is 29.95%, and thus our false negative rate is 70.05%."

* A false negative is only a problem when you include sites like it in the denominator of rate calculations - what this means to me is that you can only view 30% of the genome- I assume that this accounted for in rate calculations

Response: yes, it is accounted for in the rate calculations.

* Was the number of missed mutations even across branches? I assume quality and coverage vary by sample but also the quality of the genome might systematically obscure mutations across all samples.

Response: we have added a section (section 2) to the supplementary material to address this comment. In brief, the number of mutations was relatively constant across branches. The false negative rates range from 65.0% to 76.6%, and just one value (the highest) is more than two standard deviations away from the mean of 70.05%. The variation among branches (at least, those that lead directly to tips) seems to be almost entirely explained by variations in coverage among samples. Supplementary table 2.1 provides the full list of values, and section 2 of the supplementary information provides all the details of the coverage and correlation as well.

How are replicates treated in the analysis - if you call each replicate genotype separately it may be harder to make high quality heterozygous calls with depth ~10x/replicate. However, if the replicates are combined the depth increases to 30X and may allow more mutations to be identified. Which method was used?

Response: we use both methods, and have clarified the manuscript to make this clear. We use the first method in the positive control analysis (simply requiring that all three genotypes agree for a site to be included in downstream analyses). We use the second method (i.e. a full maximum likelihood approach that properly accounts for the likelihood of observing the three replicate read sets under the assumption that they were generated from a single underlying genotype). We have edited the manuscript throughout to try to make it clear that we have used both methods, and why. For example, the first introduction of the DeNovoGear method now reads (starting on line 118):

"This method improves on the approach we used in our positive control, above, because it increases our power to detect true somatic mutations and avoid false positives by assuming that the phylogenetic structure of the samples follows the physical structure of the tree, an assumption that is validated by the analyses above. It also makes better use of the replicate sampling design than the method we use for our positive control, above, by directly modelling the expected variation in sequencing data across our three biological replicates under the

expectation that all three replicates were sequenced from a single underlying genotype (see methods and supplementary information).”

We have also included an in-depth analysis of the replicate filtering, to answer a question raised by another reviewer. This analysis looks at the effects of requiring all three replicates to agree, or having just two, or a single replicate, on the number of mutations inferred, the false negative rate, and the false discovery rate. It is presented in section 1 of the supplementary information.

If you found 90 mutations what are the “99” high confidence mutations used to build the tree. Were some of these extra 9 false positives and could they be the reason that there are multiple most parsimonious trees?

Response: hopefully this point is now clarified in the manuscript by our attempts to describe the two different analyses that we did (see previous response). To answer the question: the 99 mutations come from the first analysis where we use a crude replicate filter in order to do a positive control and check the structure of the phylogeny matches that of the tree. The 90 mutations come from the full ML model in DeNovoGear, where we leverage some extra power by assuming that the samples follow the structure of the tree.

The first section of the supplementary information now confirms that roughly half of the 99 mutations in the first set are likely to be false positives. It may well be that this is the reason there are multiple most parsimonious trees, though that could also be the result of simply having a rather small dataset.

Is cell density similar across tissues? Presumably the rate should be correlated with cell number not size. I know it’s not likely possible to analyze this explicitly but perhaps it could be discussed

Response: thanks for the suggestion. We agree that this is an interesting question. We were (as expected, perhaps) unable to find any useful information on cell density within the tree. Given that this would mean any discussion would be very speculative (and that we don’t anticipate a lot of cell density measures being made soon), and that the paper is already rather long, we decided not to discuss this in the manuscript. Nevertheless, we do now include a related discussion towards the end of the paper, which discusses cell replication more directly (starting on line 231):

“Longevity and size are two aspects of plant life history likely to be of central importance to the evolution of somatic mutation rates. Larger plants may suffer from a higher accumulation of somatic mutations because of the necessity for additional cell divisions. Plants that live longer may suffer from a higher accumulation of somatic mutations because of the accumulation of DNA damage over time, and/or increased cell turnover in long-lived tissues. The relative importance of these two factors may differ among clades, species, and individual tissues, and is likely to also depend on the balance between DNA damage and repair between cell divisions³⁶, the accuracy of DNA replication, cell size, and the rate of cell division.”

149 - The number of mutations inherited by any one seed would not be the sum of the mutations across all branches. This means that each branch more or less should have their own mutations. The calculation of per generation mutation rate seems complex. The authors mention "with some additional assumptions" - can they expand on these assumptions and how on this per generation calculation is arrived at?

Response: We appreciate the suggestion to clarify our calculations, which we have now spelled out in more detail. The first half of the paragraph (starting line 189) now reads:

"With some additional assumptions it is also possible to estimate the mutation rate per generation, and to compare this to estimates from other plants. The average height of an adult *E. melliodora* individual is between 15 m and 30 m²⁸, so if we assume that all somatic mutations are potentially heritable (about which there is limited evidence¹ and ongoing discussion²⁹) we can estimate that the per-generation mutation rate. To do this, we assume that a typical seed will be produced from a branch that has experienced 15-30 m of linear growth from the seed²⁸, and that mutations will have accumulated along that branch at 2.75×10^{-9} somatic mutations per base per meter of physical branch length, estimated above. We therefore estimate that the heritable somatic mutation rate per generation is between 4.13×10^{-8} and 8.25×10^{-8} mutations per base."

L162: This sentence is quite confusing and the end seems to contradict the beginning: "roughly an order of magnitude lower than our estimate for *E. melliodora*, at around 3.5×10^{-10} (2.7×10^{-8} divided by the average height of individual Sitka spruce studied of 76 m) which is strikingly similar to the rate that we estimate here

Response: thanks for pointing out the lack of clarity here. We have rewritten that sentence to read (starting on line 209):

"While this per-generation rate is very similar to the one we estimate here, the rate of somatic mutation per meter of growth is around an order of magnitude lower in the Sitka spruce than our estimate for *E. melliodora*"

L163: How much does the height of a tree matter - isn't it the branching lengths that matter?

Response: We have clarified this section in the manuscript to read (starting on line 212):
"...versus 3.5×10^{-10} somatic mutations per base pair per meter of growth for Sitka spruce, estimated by dividing the per-generation mutation rate of 2.7×10^{-8} mutations per base by the average height of studied individuals of 76 m¹⁵, an appropriate calculation because the somatic mutation rate was estimated from paired samples taken from the base and the top of a collection of individual trees".

To expand on that slightly - the Sitka spruce study took samples from the base and the top of each tree (see figure pasted below). The height is thus the correct divisor to calculate the per m mutation rate, but as the reviewer points out, this was not clear from what we had written.

Figure 1A from Hanlon et al 2019.

L177: low rates of somatic mutation in large plants - are the rates lower than in small herbaceous or annual plants?

Response: yes, although the data on this for small plants remains practically limited to Arabidopsis. This is discussed on lines 198-205.

Methods

L246: "9,679,544 potential variable sites" - this is a lot given you were removing these variants with the iterative mapping method

Response: we failed to mention that this number includes all of the heterozygous genotypes called across the genome (even those with no mutation), i.e. it is simply the output of the initial VCF from GATK. The vast majority of these are then removed when we filter to only consider sites where at least one branch has a genotype that differs from those on other branches. We have now clarified this in the manuscript to read (line 300): "This resulted in a full genome alignment of all 24 samples (three replicates of eight branches), and produced an initial set of 9,679,544 potential variable sites, a number which includes all heterozygous sites in the genome."

L251: "the depth is less than or equal to 500, since excessive depth is a signal of alignment issues" Is this total depth? I.e. 500/24 ~20x depth for a sample? Normally people use something like 2x each sample's mean coverage.

Response: yes, it is the total depth. We have clarified this in the manuscript to read (line 307): "(iv) the total depth across all samples is less than or equal to 500 (i.e. roughly twice the expected depth of 240x)..."

252 "(v) the ExcessHet annotation was less than or equal to 40, since excessive heterozygosity at a site is a sign of genotyping errors" - This doesn't make sense because data were from one tree and a heterozygous site in the plant's genome should be heterozygous throughout the tree unless gene conversion or mutation changes it. Therefore all heterozygous sites will look to be in excess. You should check how big of a problem this is (ie how many mutations did it exclude)

Response: The original motivation for this (which we explained poorly in the manuscript and have now clarified on line 308) was precisely to exclude certain heterozygous sites. Manual observation of the data suggested that we frequently saw false-positive mutation calls at sites which were actually uniformly heterozygous through the tree, but at which genotyping errors had caused a mutation to be called. Using the ExcessHet filter is a convenient way to filter out such sites.

To investigate the effect of removing this filter, as suggested by the reviewer, we repeated our analysis without this filter. This re-analysis is described in section 3 of the supplementary information. As expected, removing the filter increases the number of mutations detected to 520, suggesting that applying this filter excluded 430 putative mutations, of which ~55% are false positives. Given that excluding the filter increases the false positive rate from 0.12% to 55.2%, and that this would make many downstream analyses difficult because one could not be sure if a given mutation was of biological or technical origin, we kept the filter in the main analysis. We explain this in section 3 of the supplementary information.

Journal Name: Proceedings of the Royal Society B
Journal Code: RSPB
Print ISSN: 0962-8452

Online ISSN: 1471-2954

Journal Admin Email: proceedingsb@royalsociety.org

MS Reference Number: RSPB-2019-2364

Article Status: SUBMITTED

MS Dryad ID: RSPB-2019-2364

MS Title: A phylogenomic approach reveals a low somatic mutation rate in a long-lived plant

MS Authors: Orr, Adam; Padovan, Amanda; Kainer, David; Kulheim, Carsten; Bromham, Lindell; Bustos-Segura, Carlos; Foley, William; Haff, Tonya; Hsieh, Ji-Fan; Morales-Suarez, Alejandro; Cartwright, Reed; Lanfear, Robert

Contact Author: Robert Lanfear

Contact Author Email: rob.lanfear@gmail.com

Contact Author Address 1:

Contact Author Address 2:

Contact Author Address 3:

Contact Author City: Canberra

Contact Author State: Australian Capital Territory

Contact Author Country: Australia

Contact Author ZIP/Postal Code: 0200

Keywords: somatic mutation, bioinformatics, plants, mutation rate

Abstract: Somatic mutations can have important effects on the life history, ecology, and evolution of plants, but the rate at which they accumulate is poorly understood, and has been very difficult to measure directly. Here, we demonstrate a novel method to measure somatic mutations in individual plants and use this approach to estimate the somatic mutation rate in a large, long-lived, phenotypically mosaic *Eucalyptus melliodora* tree. Despite being 100 times larger than *Arabidopsis*, this tree has a per-generation mutation rate only ten times greater, which suggests that this species may have evolved mechanisms to reduce the mutation rate per unit of growth. This adds to a growing body of evidence that illuminates the correlated evolutionary shifts in mutation rate and life history in plants.

EndDryadContent